# Improving the High-Frequency Response of PEI-Based Earphone with Sodium Copper Chlorophyllin

**DOI:** 10.3390/molecules25010219

**Published:** 2020-01-05

**Authors:** Hao-Zhi Li, Jun-Jie Wu, Wei-Jen Lee, Chien-Sheng Chen

**Affiliations:** Department of Chemistry, Fu-Jen Catholic University, New Taipei City 24205, Taiwan; jasonhemlot@gmail.com (H.-Z.L.); aab6918@gmail.com (J.-J.W.); 405176252@mail.fju.edu.tw (W.-J.L.)

**Keywords:** sodium copper chlorophyllin, porphyrin, isothermal titration calorimetry, acoustics, sound pressure level

## Abstract

The polyetherimide diaphragm, sodium copper chlorophyllin (SCC), and copper ion coating composite used on earphones were observed to improve the high-frequency (10k–14k Hz) performance. This reinforcement phenomenon was expected to make the sound experience brighter and more diverse. By SEM observation, the mixed coating of SCC/Cu^2+^ on the polyethylenimine (PEI) diaphragm exhibited a planar blocky structure and was tightly bonded to the surface of the PEI polymer without the aid of colloids. The endothermic process of SCC and metal ion complexation was analyzed by isothermal titration calorimetry. The association ratios of SCC/Cu^2+^ and SCC/Ni^2+^ were 4/1 and 6/1, respectively, and the SCC/Cu^2+^ association yielded a stronger binding constant and more free energy. It was expected that the SCC/Cu^2+^(4/1) mixed liquid would be immobilized on the PEI polymer by multivalent interaction, including hydrogen-bonding networks between carboxyl groups of SCC and amine groups of PEI, and cross-linking of bridging copper ions. We used dimethylethylenediamine (DME) monomer instead of PEI polymer to analyze this multivalent interaction and observed a two-stage exothermic association of SCC/Cu^2+^(4/1) and DME with a total Gibbs free energy of 15.15 kcal/mol. We observed that the binding energy could be used to explain that the SCC/Cu^2+^ mixed formulation could be fixed on the surface of the PEI polymer and could enhance the strength of the PEI film. Compared with graphene films, which can continuously improve the performance of high and ultrasonic frequencies, this study was devoted to and was initiated for the purpose of applying porphyrin compounds to improve music performance.

## 1. Introduction

Sound transmission and verbal expression are the bases of communication [1], and the generation of sound has been explored for thousands of years. Birds and humans use muscle tension of the glottis to generate air pressure through channels to produce different sounds [2]. Male crickets use the muscles to quickly retract the vibrating membrane [3], and then make a loud sound through the abdomen, which acts as a resonator, to attract females. The chirping of cicada is produced by the edge of the blade of one wing rubbing against the ribbed edge of the other wing [4,5,6]. The thin wing film of the cicada has a relatively large surface area, allowing it to convert the muscle contraction energy agitating the wing membrane to sound by disturbing the air, which is extremely effective in achieving sound energy conversion. Most commercial speakers and headphones reproduce sound through mechanical diaphragms, while the flat frequency response of human audibility produces a constant sound pressure level (SPL) over a frequency range of 20 to 20k Hz [7,8,9]. The diaphragm of the loudspeaker oscillates during its operation to form a basic harmonic oscillator with inherent mass, restoring force, spring constant, and damping [10,11]. Most insect or instrument resonators exhibit a slightly dampened sharp frequency response; in contrast, speakers producing wideband audio typically require significant damping to increase the response. An ideal audio conversion diaphragm should have a small mass and a soft spring constant, with high power efficiency that facilitates conversion of most of the input energy into sound waves. For small headphones, a thinner and lower mass density diaphragm is needed to maintain the advantages of air damping. 

Graphene has an extremely low mass density and high mechanical strength, making it an ideal composite for high efficiency and high quality wideband audio speakers [12]. Graphene has been used to construct mechanical resonators such as graphene thermoacoustic devices [13,14,15], electrostatic graphene speakers [16,17], graphene/PVDF composite films [18,19], graphene/paper/ PCB source devices [13], etc. Given the fact that graphene is a flat monolayer or has multiple layers of carbon atoms that are tightly packed into a two-dimensional honeycomb lattice, we developed a composite acoustic film of sodium copper chlorophyllin (SCC) and polyethylenimine (PEI) film and hypothesized that an SCC with a porphyrin core structure would produce an excellent frequency response throughout the audible area.

## 2. Result and Discussion

Sodium copper chlorophyllin is widely used in the food industry, and this tetrapyrrole compound is relatively easy to obtain and manufacture in large quantities [20]. Having three carboxyl functional groups, SCC has good water solubility but is poorly soluble in organic solvents. The use of metal ions to sequester SCC might facilitate the concealment of polar functional groups and enhance the chance of dissolution of the metal ion chelating complex in methyl propylene glycol ether (MPG, Appendix A). The complexations between SCC and metal ions were analyzed by calorimetric measurement, namely isothermal titration calorimetry (ITC, Figure 1, Table 1, and Appendix A). The binding constant of Cu^2+^ ions to SCC was 2.22 × 10^5^ M^−1^, and the complexation ability of Ni^2+^/SCC was 9.01 × 10^4^ M^−1^. We observed that the binding isotherms between metal ions and SCC were endothermic processes, while the enthalpies (Δ*H*) of Cu^2+^/SCC and Ni^2+^/SCC were 0.33 and 1.03 kcal/mol, respectively. Large entropy (Δ*S*) was observed in the binding process of Cu^2+^/SCC and Ni^2+^/SCC, where the determining values were 25.6 and 26.1 cal/mol/K, respectively, and the free energies were −7.30 and −6.75 kcal/mol, respectively. A more stable complex formation is often accompanied by increased entropy since the binding process of metal ions with ligands tends to prevent the binding structure from initiating solvating network. This kind of thermodynamic phenomenon where the binding process simultaneously displaced the solvation group was similarly reported in EDTA/Ca^2+^ titration [21]. Compared to Zn^2+^ or Ca^2+^, we concluded that the stable complexation of SCC/Cu^2+^(4/1) and SCC/Ni^2+^(6/1) could displace the relative ions of the individual groups and thermodynamically form a stable complex.

We then developed a simple liquid coating method to construct an SCC/M^2+^ composite thin layer on the acoustic diaphragm of the PEI film and tested the acoustic response. According to our understanding, this is the first attempt to use an SCC/M^2+^ composite film as a diaphragm. In addition, porphyrin containing conjugates were studied as microbubble contrast agents [22,23] and found to be intrinsically suitable for both ultrasound and photoacoustic imaging. The SCC/Cu^2+^(4/1) and SCC/Ni^2+^(6/1) complexes were each mixed and dissolved in MPG. Figure 2A,B depicts the acoustic response of a PEI film speaker coating with different amounts of SCC/Cu^2+^(4/1) complex or SCC/Ni^2+^(6/1) complex compared to a native PEI film. Our approach was to stack and perform acoustic testing on the same single PEI earphone. The high-frequency (>10k Hz) frequency response increased with the increase of SCC/M^2+^ composites, especially in the 10k–14k Hz range, which meant that these SCC/Cu^2+^/PEI composite diaphragms made the high-frequency sound more beautiful and rich. When the stack weight reached 600 and 800 ppm (SCCcomplex/PEI, *w*/*w*), the high-frequency sound responses at 13.3k Hz were 2.5 dB and 2.7 dB higher than the native PEI diaphragm, respectively (Figure 2C,D). Compared with the SCC/Cu^2+^ composite PEI diaphragm, the acoustic response of the high-frequency 10k–14k Hz could be improved. The SCC/Ni^2+^ (800 ppm) composite film improved the spectral response of 10.4k Hz by 2.0 dB; at the same amount, the SCC/Ni^2+^ (800 ppm) composite diaphragm of the spectral response at 5.8k Hz and 13.3k Hz was suppressed to 0.8 and 0.2 dB, respectively. This result means that the acoustic performance of the SCC/Cu^2+^/PEI diaphragm was superior to that of the SCC/Ni^2+^ composite PEI film, and this result led us to continue our examination of copper ion cross-linked SCC and PEI polymers.

We also attempted to coat the same composite formulation of SCC/Cu^2+^ on a polyethylene terephthalate (PET) polymer diaphragm, which is more commonly used in commercial earphones. However, the SCC/Cu^2+^ complex did not adhere to the PET polymer coating and thus did not enhance the acoustic response. In order to verify that the SCC/Cu^2+^ complex could be bonded to the PEI diaphragm by a metal ion bridge, we planned to simulate the PEI polymer with dimethylethylenediamine (DME) and applied ITC to analyze the association between SCC/Cu^2+^ and DME (Figure 3). As seen in Table 2 (entry 1,2), DME molecules could chelate with copper ions and sodium copper chlorophyllin, and the binding Gibbs free energies (Δ*G*) were −5.54 and −5.65 kcal/mol, respectively. Among them, the DME/Cu^2+^ binding process had higher enthalpy (−9.45 kcal/mol) and entropy (−13.10 cal/mol/K). We also observed that the DME and SCC/Cu^2+^(4/1) binding process was a two-stage exothermic binding process, and the binding ratios (n1 and n2) were 1 and 4, as seen in Figure 3C and Table 2 (entry 3). The first stage of the binding enthalpy was −9.23 kcal/mol, which was close to the DME/Cu^2+^ association enthalpy value (−9.45 kcal/mol, entry 1), and this value was applied to explain that DME could chelate the central copper ion of the SCC/Cu^2+^ complex (Figure 4A). It is worth mentioning that the association process at the first stage had a larger equilibrium constant (*K*_a_, 9.42 × 10^6^ M^−1^) and a small entropy (−0.95 cal/mol/K), and the DME bonding process at this stage could not significantly change the surrounding structure of the ionic SCC/Cu^2+^(4/1) structure. Conversely, the second stage of additional DME combined with SCC/Cu^2+^ had an increased entropy value (7.10 cal/mol/K), which might indicate that the surrounding counterions, i.e., sodium ions, of the carboxylate groups of SCC/Cu^2+^ complex could be squeezed out and the disorder of the association structure could be increased. In summary, this two-stage combination provided Δ*G* values of −8.95 and −6.20 kcal/mol, respectively. Those thermodynamic values of SCC/Cu^2+^/DME provided 15.15 kcal/mol of free energy and strongly indicated, indirectly, that SCC/Cu^2+^ could be used as a coating on PEI polymers by multivalent interaction (Figure 4B).

Since sodium copper chlorophyllin has low solubility and diamagnetic copper ions, the complexation of SCC with Cu^2+^ cannot be identified by liquid nuclear magnetic resonance spectroscopy. We note that SCC has one 1-carboxyl group that can participate in the aromatic resonance and two non-aromatic carboxylates at the 3- and 5-positions. We planned to use sodium benzoate (SB) with aromatic carboxyl groups and sodium citrate (SC) with non-aromatic carboxyl groups as control experiments and to use FT-IR spectrum identification to help explain that the aromatic 1-carboxylate of SCC could interact with Cu^2+^ (Table 3 and Appendix A). Compared to the carboxyl vibration of SCC (1568 cm^−1^) and SB (1588 cm^−1^), the mixing of copper ions with SCC and SB showed that the stretching vibrations were 1554 and 1595 cm^−1^, respectively. The carboxylate group of SC at 1596 cm^−1^ was observed with the same wavenumber with or without copper ions, indicating that the conjugated 1-carboxylate of SCC might lead an important role in the association of copper ions. Additionally, the stretching vibration for the carboxylates of the SCC/Cu^2+^/DME(4/1/4) and SC/Cu^2+^/DME(3/1/3) were 1571 and 1567 cm^−1^, respectively, and such similar values suggested that the 3- and 5-carboxylate groups of SCC could have hydrogen-bonding networks with the amino groups of DME. Otherwise, the addition of DME into the SB/Cu^2+^ mixture did not influence the original stretching wavenumbers (1595 and 1550 cm^−1^) of the chelating SB/Cu^2+^ complex, and this significant observation supports the notion that the 1-carboxylate of SCC might be responsible for the chelation of copper ions. Taking the EDTA/Cu^2+^ complexation as the template (Figure 4), copper ions with hexacoordinate character properties can chelate with two amino groups and four carboxyl groups of SCCs. From the thermodynamic analysis of ITC (Table 2, entry 2,3), it was determined that four SCCs could form a complex with four DME molecules. The optimal binding ratio for SCC/Cu^2+^ to DME was 4; however, this complexation structure was expected to contain a central copper ion that could chelate with one DME molecule, since SCC/Cu^2+^/DME could form a complex structure at a ratio of 4/1/4 and the internal steric hindrance formed by the more crowded structure might limit the combination of SCC/Cu^2+^ with more DME.

Conventional techniques for composite coatings for polymeric diaphragms, such as colloidal bonding, are based on the application of a specific polymer, and uniform film formation is indispensable. In this regard, it is of interest to note whether SCC/Cu^2+^ coatings form a film on PEI polymers without the binder. The topography of the coated SCC/Cu^2+^ composite on PEI film was assessed based on SEM images. The images showed that the composite coating appeared as a blocky structure, and the length and thickness of this blocky accumulation were about 8 and 0.5 μm, respectively, as seen in Figure 5A. As described in the previous paragraph, the SCC/Cu^2+^ composite could be combined with the PEI film by multivalent interaction to cohere into a large-area blocky structure after it dried, and this blocky material fine-tuned the acoustic qualities of the diaphragm vibration. Compared with graphene composite diaphragms, the performance of treble [16,18] and ultrasonic sound [17] can be continuously enhanced because these nano-materials can form a continuous and large-area multilayer on the base polymer, and their use as a reinforcement in polymer composites has huge potential for further enhancement of the mechanical properties of composites. The SCC/Cu^2+^/PEI composite diaphragm could only improve the performance of local high-frequencies (10k–14k Hz). The possible reason for this might be that the SCC/Cu^2+^ complex could not form a continuous and uninterrupted planar film on the PEI film to enhance the rigidity of the composite film. This phenomenon can be demonstrated by the tensile elasticity of the PEI composite; Young’s modulus of the 600 ppm SCC/Cu^2+^/PEI composite film was 1.08 GPa (Table 4), which was 0.12 Gpa higher than that of the standard PEI diaphragm. Relative to the two-dimensional plane formed by the covalent carbon-carbon bond of graphene, the SCC/Cu^2+^ complexes constructed by the four SCCs with the copper ions were cooperatively linked by the metal bridge. The associated Gibbs free energy of SCC/Cu^2+^ complexation was 7.3 kcal/mol, close to the two hydrogen bond energies, and still much less than the carbon–carbon covalent bond energy. In terms of the implications for acoustics, the SCC/Cu^2+^/PEI composite diaphragm could transmit sound characteristics different from the graphene composite speaker. The enhancement of the high-frequency band (10k–14k Hz) in this study expanded the purity and spatial sense of the sound, for example, by increasing the crisp sound of the guitar [24], beautifying the high-frequency performance of the percussion [25], and improving the comprehensive sound of the symphony and the electronic music [26]. We noticed that in the analysis of the classic Stradivari violin, the content of certain inorganic elements is not naturally increased; for example, Cu^2+^, Ca^2+^, and Al^3+^, inferred bivalent and trivalent metal ions, may make cellulose and lignin cross-link together. These metal cross-links seem to change the sound quality of the Stradivari violin [27,28,29]. From our results, the divalent copper ion could cross-link the SCC and PEI together and improve the treble-sound quality, and the analytic system developed here might have potential to approximate the metal ion cross-linking effect with lignin/cellulose-based earphones.

## 3. Experimental Section

### 3.1. General Information

Sodium copper chlorophyllin [SCC], copper perchlorate [Cu(ClO_4_)_2_], nickel perchlorate [Ni(ClO_4_)_2_], zinc perchlorate [Zn(ClO_4_)_2_], and calcium perchlorate [Ca(ClO_4_)_2_] were purchased from Sigma-Aldrich Chemical Co. (St. Louis, MO., USA). Methyl propylene glycol ether was purchased from Merck Chemical Co. (Darmstadt, Germany). PEI-diaphragm earphone (φ, 1 cm, 10 Ω) and PEI membrane were purchased from Yuan Zhi Li Acoustic Co. (Dongguan, China). FT-IR absorbance was determined with a PerkinElmer Spectrum 100 (PerkinElmer, Massachusetts, MA, USA). FE-SEM was recorded with a JEOL JSM-7800F Prime Microscope (JEOL, Tokyo, Japan) using Schottky field emission. Young’s modulus measurement was detected with a TA.XT*plusC* Texture Analyzer (Satble Micro System, Surrey, UK) using standard tension method.

### 3.2. Isothermal Titration Calorimetry of SCC/Cu^2+^, SCC/Ni^2+^, SCC/Zn^2+^, SCC/Ca^2+^, Cu^2+^/DME, SCC/DME, [SCC/Cu^2+^]/DME Mixtures

ITC measurements were conducted on a VP-ITC Microcalorimeter (Microcal Inc., Northampton, MA, USA). The heat produced by the complex formation between SCC (1.436 mL in H_2_O) and metal ion at 25 °C with continuous stirring (300 rpm) was analyzed using Origin software. The integration of the heat pulses obtained from each titration was fitted to a standard binding curve to obtain the enthalpy change (Δ*H* in cal/mol), the entropy change (Δ*S* in cal mol^−1^ K^−1^), the association constant (Kassoc in M^−1^), and the number of binding sites (n). The integration of the heat pulses obtained from each titration was fitted to a two-site binding curve for SCC/Cu^2+^ and [SCC/Cu^2+^]/DME, and raw dates obtained from titrations of SCC/Ni^2+^, SCC/Zn^2+^, SCC/Ca^2+^ were fitting with one-site binding curve.

### 3.3. SCC/Metal Coating on PEI Diaphragm and Frequency Response Measurement

The coating solution was prepared by a mixture of sodium copper chlorophyllin (2 mg, 82.1 μmol) and metal perchlorate (20.5 μmol) in Methyl propylene glycol ether (1 mL). This SCC/metal complex solution (5 μL/time) was coated to the PEI diaphragm (φ, 1 cm) and dried in an oven (60 °C) for 30 min. The acoustic platform for testing the SCC/M^2+^/PEI-composite sound source containing a signal generator, a standard microphone, an AE-2 ear simulator kit (IEA Electro-Acoustic Tech., Taipei, Taiwan), and a dynamic signal analyzer. The standard microphone (1/4 inch, GRAS, Holte, Denmark), which had a flat frequency response reaching from 10 Hz to 21k Hz, was used to measure the sound pressure level (SPL) of the loudspeaker sound source. EA-2 electro-acoustic integrated system (IEA Electro-Acoustic Tech, Taipei, Taiwan) and the signal analyze software (CLIO 12.0, Firenze, Italy) was used to generate a sound signal and record the value of sound pressure level.

## 4. Conclusions

In this study, we provided a quantitative assessment of the complexation of sodium copper chlorophyllin with copper ions and dimethylethylenediamine using a combination of ITC, infrared spectroscopy, and SEM. Through calorimetric measurements, we observed the binding of SCC/M^2+^ linked by metal bridging, including the selective association of SCC/Cu^2+^ with a binding ratio of 4/1, and SCC/Ni^2+^ with a binding ratio of 6/1, which was compared with the stronger association constant (2.22 × 10^5^ M^−1^) for the titration of SCC and copper ions. At the same time, we also observed the SCC/Cu^2+^ composite topography on a PEI diaphragm and concluded that the SCC/Cu^2+^ complex could be bound to the PEI polymer by multivalent interaction, which might be indirectly attributed to the free energy of 15.15 kcal/mol for the complexation of SCC/Cu^2+^(4/1) and DME. The SCC/Cu^2+^ complex and the PEI polymer were bound by copper ions cross-linking and polar hydrogen bonding, which enhanced the high-frequency sound responses (10k–14k Hz) of the PEI based composite and was expected to make the sound more beautiful and complex. It is known from the literature that SCC is an organic compound that becomes unstable after prolonged heat treatment [30], which is disadvantageous for application of the developing wet coating method in this study for the formation of large-scale acoustic films. This led us to continue to synthesize thermostable porphyrin derivative with multiple carboxylates, and to develop a mass production system that could be done by inkjet printing or roll-to-roll coating.

## Figures and Tables

**Figure 1 molecules-25-00219-f001:**
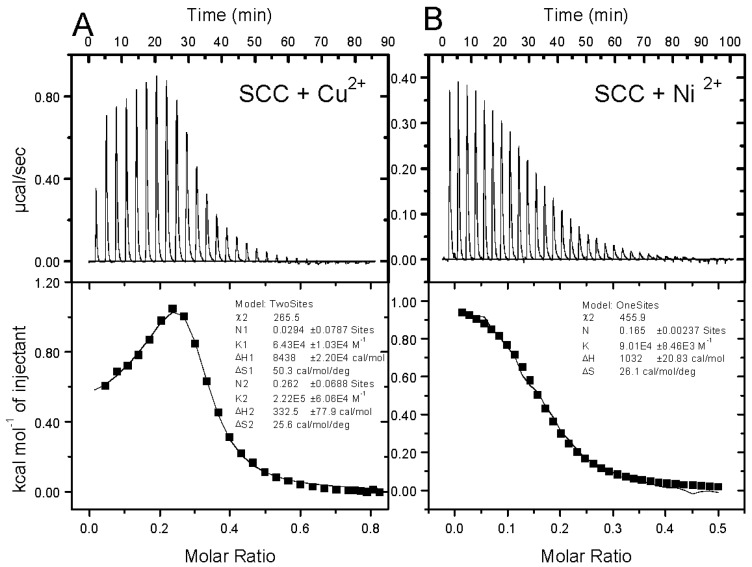
Binding isotherms for sodium copper chlorophyllin (SCC) titrated with copper and nickel ions. Titrations of (**A**) 0.76 mM SCC with 4.2 mM Cu(ClO_4_)_2_ and (**B**) 0.76 mM SCC with 1.89 mM Ni(ClO_4_)_2_ in H_2_O at 25 °C were performed using an isothermal titration calorimetry (ITC) microcalorimeter. The integrated fitted curves show the experimental points with a sequential binding site function for SCC/Cu(ClO_4_)_2_ titration and with a one-site function for SCC/Ni(ClO_4_)_2_ titration.

**Figure 2 molecules-25-00219-f002:**
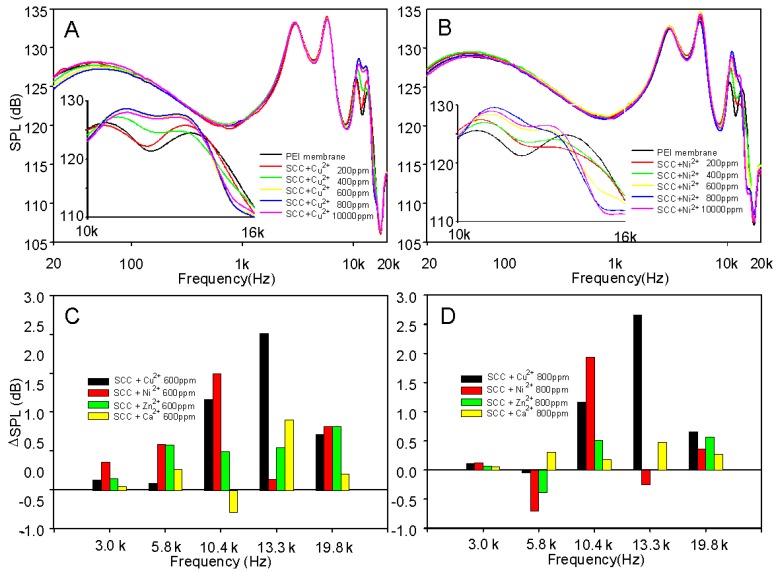
Frequency response curves of PEI-earphone diaphragms made of SCC/Cu^2+^ (**A**) and SCC/Ni^2+^ (**B**) composites. Differences of sound pressure level, including SCC/Cu^2+^, SCC/Ni^2+^, SCC/Zn^2+^, and SCC/Ca^2+^ composites on PEI diaphragms at 600 (**C**) and 800 (**D**) ppm. ΔSPL = [SPL]_composites_—[SPL]_blank_.

**Figure 3 molecules-25-00219-f003:**
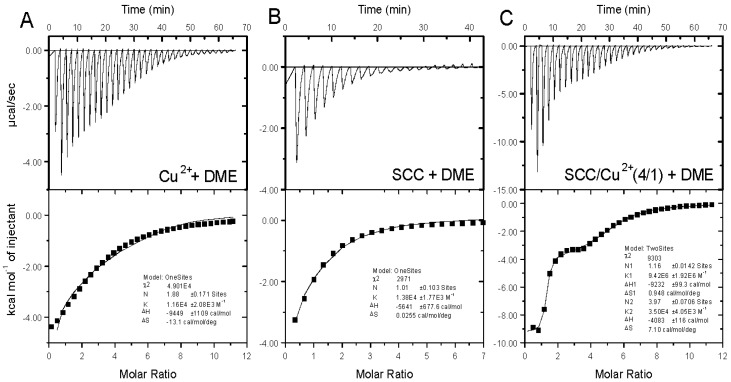
Binding isotherms for complexation of Cu^2+^, SCC, and SCC/Cu^2+^ with DME at 25 °C. Titrations of (**A**) 84.3 μM Cu(ClO_4_)_2_ with 5.0 mM DME, (**B**) 84.3 μM SCC with 5.0 mM DME, and (**C**) 84.3 μM SCC/Cu^2+^(4/1) with 5.0 mM DME in H_2_O at 25 °C were performed using an ITC microcalorimeter.

**Figure 4 molecules-25-00219-f004:**
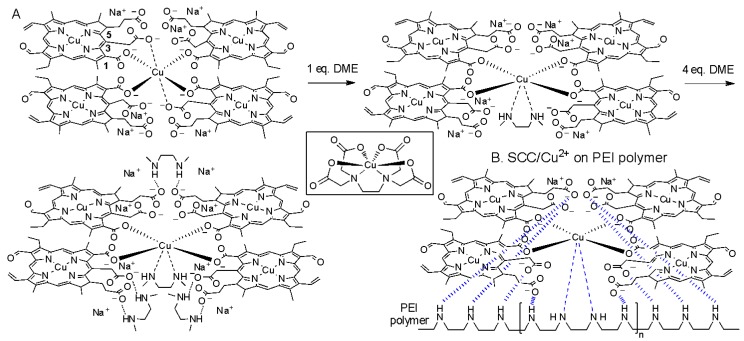
Proposed binding model for SCC/Cu^2+^ with DME (**A**) and SCC/Cu^2+^ on PEI polymer (**B**). Inset: structure for the complexation of EDTA/Cu^2+^.

**Figure 5 molecules-25-00219-f005:**
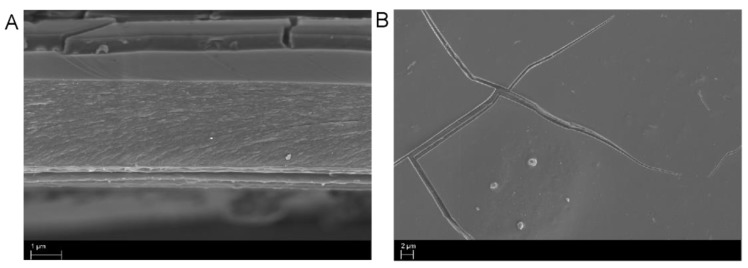
Scanning electron microscopy (SEM) images of coated SCC/Cu^2+^(4/1) on PEI film at 10,000× (**A**, side view) and 2000× (**B**, top view) magnification.

**Table 1 molecules-25-00219-t001:** Thermodynamic parameter of SCC/M^2+^ complexation in water at 25 °C.

Entry	M^2+^	1/n (SCC/M^2+^)	*K*_a_/10^4^ (M^−1^)	Δ*H* (kcal/mol)	Δ*S* (cal/mol/K)	Δ*G* (kcal/mol)
1	Cu^2+^	3.82	22.2	0.33	25.6	−7.30
2	Ni^2+^	6.06	9.01	1.03	26.1	−6.75
3 ^a^	Zn^2+^	-	-	-	-	-
4 ^a^	Ca^2+^	-	-	-	-	-

^a^ Significant binding isotherm was not observed.

**Table 2 molecules-25-00219-t002:** Thermodynamic parameter for complexation of Cu^2+^, SCC, and SCC/Cu^2+^ with DME at 25 °C.

Entry	Complex	n (Complex/DME)	*K*_a_/10^4^ (M^−1^)	Δ*H* (kcal/mol)	Δ*S* (cal/mol/K)	Δ*G* (kcal/mol)
1 ^a^	Cu^2+^	1.88	1.16	−9.45	−13.10	−5.54
2 ^b^	SCC	1.01	1.38	−5.64	0.03	−5.65
3 ^c^	SCC/Cu^2+^ (4/1)	1.16 (n1) 3.97 (n2)	942 3.50	−9.23 −4.08	−0.95 7.10	−8.95 −6.20

^a^ Raw data obtained from titration of Cu^2+^ (84.3 μM) with automatic injections (8 μL each) of DME (5.0 mM). The integrated fitting curve showing the experimental points by using a one-site binding fitting function. ^b^ Raw data obtained from titration of SCC (84.3 μM) with automatic injections (8 μL each) of DME (5.0 mM). The integrated fitting curve showing the experimental points by using a one-site binding fitting function. ^c^ Raw data obtained from titration of SCC (84.3 μM) and Cu^2+^ (28.1 μM) with automatic injections (8 μL each) of DME (5.0 mM). The integrated fitting curve showing the experimental points by using a two-site binding fitting function.

**Table 3 molecules-25-00219-t003:** FT-IR vibration wavenumbers (cm^−1^) for the carboxylate groups of SCC, SB, and SC with or without Cu(ClO_4_)_2_ and DME ^a^.

Entry	Sample	*δ*_sample_ without Cu^2+^	*δ*_sample_ with Cu^2+^	*δ*_sample_ with Cu^2+^+DME
1	SCC	1568(S)	1554(M)	1571(M) ^b^
2	sodium benzoate (SB)	1588(M), 1552(S)	1595(M), 1550(S)	1595(M), 1550(S) ^c^
3	sodium citrate (SC)	1596(M), 1554(S)	1596(M), 1557(M)	1567(M) ^d^

^a^ Coating on KBr salt tablets. S, strong; M, medium; W, weak. ^b^ SCC/Cu^2+^/DME = 4/1/4. ^c^ SB/Cu^2+^/DME = 2/1/2. ^d^ SC/Cu^2+^/DME = 3/1/3.

**Table 4 molecules-25-00219-t004:** Young’s modulus (*E*) for SCC/Cu^2+^/PEI composite ^a^.

Entry	PEI Composite	*E* (GPa)
1	PEI polymer	0.96 ± 0.15
2	400 ppm SCC/Cu^2+^	0.97 ± 0.18
3	600 ppm SCC/Cu^2+^	1.08 ± 0.16

^a^ SCC/Cu^2+^ = 4/1.

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
