# Peer review of "Improving the High-Frequency Response of PEI-Based Earphone with Sodium Copper Chlorophyllin"

_molecules, 2020, doi:10.3390/molecules25010219_

Round 1

Reviewer 1 Report

The article is very interesting, showing an original application of the sodium copper chlorophyllin (SCC) in a very different field from that usually considered. Exploiting the SCC ability in chelating bivalent metal cations, a composite material constituted by SCC/metal ion complex and a polymer, like PEI, is proposed as coating layer for improving acoustic performances of earphone. First of all, the manuscript needs a thorough revision of the English language. There are several incomprehensible sentences, for example in the Introduction section, lines 33-34, what does it mean that "the cicada is produced..."? The cicada is an insect, so...

Regarding the  Introduction section, I don’t understand very well what cockroaches and crickets have to do with the type of work. Besides the fact that the sentence at lines 31-33 is confused.

In the first part of the Results and Discussion section, the authors state (lines 59-61) that SCC can be sequestered by metal ions, but I could not find any bibliographical reference to this claim. Should I take this as an anticipation of the results of the work? Please clarify.

At lines 95-96, what does the sentence "...a composite film speaker with different 95 numbers of SCC/Cu2+ and SCC/Ni2+ compared to a native PEI film." mean? Do you mean "containing different SCC/metal ion amounts"? Please, clarify.

Please, make clearer the choice of focusing the attention only on the composite with Cu2+ (lines 103-107).

At line 152, the authors begin to describe FTIR analysis of the SCC/Cu2+ complex in presence of DME. After, they insert sodium benzoate and sodium citrate. Why? It is absolutely unclear the role of these two sodium salts. Please, integrate the discussion.

What is meant by "...changes for the stretching vibration were -14 and 7 cm-1, respectively"?

Although the use of Conclusion title or the Conclusion section is optional, in my opinion the indication of the concluding part of the manuscript would make it easier to read.

Another aspect regards the sentences at lines 216-223. I don't understand well why, in the conclusive part, the authors mention the role of bivalent and trivalent metal ions in the sound quality of a Stradivari's violin, or the thermostability of SCC. The link with the results of the work is not clear.

Author Response

Thanks for the comments. Based on the comments given by the reviewers, we greatly modified the English expression with editor service, and the changed text was marked in yellow in the revised manuscript.

Question 1.  First of all, the manuscript needs a thorough revision of the English language. There are several incomprehensible sentences, for example in the Introduction section, lines 33-34, what does it mean that "the cicada is produced..."? The cicada is an insect, so...

Author Respond:  Author modified the English language with an English editor, and change as following in line 33-35.

“Male crickets use the muscles to quickly retract the vibrating membrane [3], and then make a loud sound through the abdomen, which acts as a resonator, to attract females.”

Question 2.  Regarding the  Introduction section, I don’t understand very well what cockroaches and crickets have to do with the type of work. Besides the fact that the sentence at lines 31-33 is confused.

Author Respond:  Author modified the English language with an English editor, and change as following in line 35-36.

“The chirping of cicada is produced by the edge of the blade of one wing rubbing against the ribbed edge of the other wing.”

Question 3.  In the first part of the Results and Discussion section, the authors state (lines 59-61) that SCC can be sequestered by metal ions, but I could not find any bibliographical reference to this claim. Should I take this as an anticipation of the results of the work? Please clarify.

Author Respond:  For the phenomenon of adding metal ions to help SCC dissolve, author added a photo to show in Supplementary Materials Figure S1 (see the last page of attachment), at the same time, modified the description and noted in line 69-71, the text is as follows.

“The use of metal ions to sequester SCC might facilitate the concealment of polar functional groups and enhance the chance of dissolution of the metal ion chelating complex in methyl propylene glycol ether (MPG, Supplementary Materials Figure S1).”

Question 4.  At lines 95-96, what does the sentence "...a composite film speaker with different 95 numbers of SCC/Cu2+ and SCC/Ni2+ compared to a native PEI film." mean? Do you mean "containing different SCC/metal ion amounts"? Please, clarify.

Author Respond:  Author modified this section as following in line 98-100.

“Figure 2A–B depicts the acoustic response of a PEI film speaker coating with different amounts of SCC/Cu2+ (4/1) complex or SCC/Ni2+ (6/1) complex compared to a native PEI film.”

Question 5.  Please, make clearer the choice of focusing the attention only on the composite with Cu2+ (lines 103-107).

Author Respond:  This is a good suggestion. Author changed as following in line 110-112.

“This result means that the acoustic performance of the SCC/Cu2+/PEI diaphragm was superior to that of the SCC/Ni2+ composite PEI film, and this result led us to continue our examination of copper ion cross-linked SCC and PEI polymers.”

Question 6.  At line 152, the authors begin to describe FTIR analysis of the SCC/Cu2+ complex in presence of DME. After, they insert sodium benzoate and sodium citrate. Why? It is absolutely unclear the role of these two sodium salts. Please, integrate the discussion.

Author Respond:  We added a section in line 158-165 as following.

“Since sodium copper chlorophyllin has low solubility and diamagnetic copper ions, the complexation of SCC with Cu2+ cannot be identified by liquid nuclear magnetic resonance spectroscopy. We note that SCC has one 1-carboxyl group that can participate in the aromatic resonance and two non-aromatic carboxylates at the 3- and 5-positions. We planned to use sodium benzoate (SB) with aromatic carboxyl groups and sodium citrate (SC) with non-aromatic carboxyl groups as control experiments and to use FT–IR spectrum identification to help explain that the aromatic 1-carboxylate of SCC could interact with Cu2+ (Table 3 and Supplementary Materials, Figure S3–S11).”

Question 7.  What is meant by "...changes for the stretching vibration were -14 and 7 cm-1, respectively"?

Author Respond: Author changed as following in line 165-167.

“Compared to the carboxyl vibration of SCC (1568 cm–1) and SB (1588 cm–1), the mixing of copper ions with SCC and SB showed that the stretching vibrations were 1554 and 1595 cm–1, respectively.”

Question 8.  Although the use of Conclusion title or the Conclusion section is optional, in my opinion the indication of the concluding part of the manuscript would make it easier to read.  Another aspect regards the sentences at lines 216-223. I don't understand well why, in the conclusive part, the authors mention the role of bivalent and trivalent metal ions in the sound quality of a Stradivari's violin, or the thermostability of SCC. The link with the results of the work is not clear.

Author Respond:  We add the title of the “conclusion” to separate the “discussion” and “conclusion”. At the same time, we moved the lines 216-223 of the original paper in to the line 216-222 of edited manuscript.

Reviewer 2 Report

The article is very interesting.
In the Results and Discussion section:In presenting the results bibliographic sources should be used. Comparison of the results with the specialized literature. For example: lines 59-61- the authors state but I could not find any bibliographical reference to this claim
The aim of the work should be better sketched and the results obtained should be highlighted.
Even if there is no section of Conclusions these should be better presented in correlation with the obtained results. For example lines 216-223.

Author Response

Thanks for the comments. Based on the comments given by the reviewers, we greatly modified the English expression with editor service, and the changed text was marked in yellow in the revised manuscript.

Question 1:  In the Results and Discussion section: In presenting the results bibliographic sources should be used. Comparison of the results with the specialized literature. For example: lines 59-61- the authors state but I could not find any bibliographical reference to this claim.

Author Respond:  For the phenomenon of adding metal ions to help SCC dissolve, author added a photo to show in Supplementary Materials Figure S1(see the last page of attachment), at the same time, modified the description and noted in line 69-71, the text is as follows.

“The use of metal ions to sequester SCC might facilitate the concealment of polar functional groups and enhance the chance of dissolution of the metal ion chelating complex in methyl propylene glycol ether (MPG, Supplementary Materials Figure S1).”

Question 2:  The aim of the work should be better sketched and the results obtained should be highlighted. Even if there is no section of Conclusions these should be better presented in correlation with the obtained results. For example lines 216-223.

Author Respond:  We add the title of the “conclusion” to separate the “discussion” and “conclusion”. At the same time, we moved the lines 216-223 of the original paper in to the line 216-222 of edited manuscript.

Round 2

Reviewer 1 Report

The new version is acceptable.